# Additive-Assisted Optimization in Morphology and Optoelectronic Properties of Inorganic Mixed Sn-Pb Halide Perovskites

**DOI:** 10.3390/ma15030899

**Published:** 2022-01-25

**Authors:** Rubaiya Murshed, Shubhra Bansal

**Affiliations:** Department of Mechanical Engineering, Howard R. Hughes College of Engineering, University of Nevada, Las Vegas, NV 89154, USA; murshed@unlv.nevada.edu

**Keywords:** perovskite films, stability, guanidinium thiocyanate, cesium chloride, Sn-Pb, DMSO, photovoltaic

## Abstract

Halide perovskite solar cells (HPSCs) are promising photovoltaic materials due to their excellent optoelectronic properties, low cost, and high efficiency. Here, we demonstrate atmospheric solution processing and stability of cesium tin-lead triiodide (CsSnPbI_3_) thin films for solar cell applications. The effect of additives, such as pyrazine and guanidinium thiocyanate (GuaSCN), on bandgap, film morphology, structure, and stability is investigated. Our results indicate the formation of a wide bandgap (>2 eV) structure with a mixed phase of tin oxide (SnO_2_) and Cs(Sn, Pb)I_3_. The addition of pyrazine decreases the intensity of SnO_2_ peaks, but the bandgap does not change much. With the addition of GuaSCN, the bandgap of the films reduces to 1.5 eV, and a dendritic structure of Cs(Sn, Pb)I_3_ is observed. GuaSCN addition also reduces the oxygen content in the films. To enable uniform film crystallization, cesium chloride (CsCl) and dimethyl sulfoxide (DMSO) additives are used in the precursor. Both CsCl and DMSO suppress dendrite formation with the latter resulting in uniform polycrystalline films with a bandgap of 1.5 eV. Heat and light soaking (HLS) stability tests at 65 °C and 1 sun for 100 h show all film types are stable with temperature but result in phase segregation with light exposure.

## 1. Introduction

HPSCs have been performing as a competitive technology for next-generation solar cells due to their high power conversion efficiency (PCE) of over 25% [1]. They offer excellent light absorption, charge-carrier mobility, and lifetime, which results in high efficiency with low-cost fabrication and scalable manufacturing technology [2]. To achieve this performance, it will require us to control the recombination at perovskites grain boundaries and enhance material stability. Different additives have been investigated to reduce bulk defect density and recombination rate by passivating the perovskite interface. A suitable organic cation can reduce water penetration in Sn-based perovskite devices [3]. Jokar et al. [4] introduced nonpolar organic Guanidinium (GA) cation in FASnI_3_ to form (GA, FA)SnI_3_. At a precursor ratio (GAI: FAI) of 20:80, in addition to 1 mol% ethylene diammonium-diiodide (EDAI_2_), they were able to reduce pinholes and enhance the uniform nucleation, and, by lowering the concentration to 15%, they have achieved significant improvement in the photoluminescence (PL) lifetime of the films, from 0.7 ns to 1.4 ns. For the overall performance of an efficient photovoltaic device, the capability of the perovskite absorber layer for the generation and extraction of charge carriers is highly required. Recombination at grain boundaries and perovskite interface with electron and hole transport layers results in non-radiative carrier loss. Suppressing recombination pathways within grain boundaries is essential for mitigating carrier loss and enhancing efficiency. It is found that the GuaSCN additive can notably improve the PCE and the stability of methylammonium lead-triiodide (MAPbI_3_) perovskite solar cells due to the improvement of quality and crystal structure of perovskite films. Marco et al. [5] observed a reduced effect of activation energy and enhanced carrier lifetime due to the passivation effect of partial GA incorporation in MAPbI_3_. Open-circuit voltage (V_oc_) is one of the vital photovoltaic parameters which impact PCE in HPSCs. High V_oc_ also accommodates non-feasible electrochemical reactions (e.g., water splitting and carbon dioxide reduction) [6,7,8]. Minimizing voltage losses can improve V_oc_, as well as the overall performance of HPSCs. Additive engineering has been used as a functional tool to remarkably enhance V_oc_ and PCE in HPSCs [9]. Apart from using GA as an additive, guanidinium based on different salts, such as guanidinium thiocyanate (GuaSCN) and guanidinium chloride (GCl), have also been explored to improve the photovoltage of HPSCs [10]. Pham et al. [11] reported that GuaSCN-treated MAPbI_3_ films have grain structure optimization from nano-scale to micron-size with a low density of grain boundaries. According to them, post-treatment by the GuaSCN solution passivates the surface of MAPbI_3_ perovskite and reduces the grain boundaries at the perovskite interface. The influence of GA^+^ dopant in controlling hysteresis and enhancing performance in mixed-cation formamidinium-cesium lead triiodide (FA_0.83_Cs_0.17_PbI_3_) was also studied [12]. Tong et al. [13] reported using GuaSCN to improve the structural and optoelectronic properties of low bandgap (1.25 eV) in Sn-Pb mixed perovskite thin films, by reducing defect densities and increasing carrier lifetime and diffusion length they demonstrated >20% of PCE. They attributed these improvements to the formation of two-dimensional (2D) structures at grain boundaries that appeared to passivate grain boundaries, suppress the formation of excessive Sn vacancies, and enhance the stability of low bandgap perovskite films. GuaSCN is expected to preferentially reside at grain boundaries and forms hydrogen bonds with under-coordinated iodine species to effectively suppress charge trapping/recombination regions [5]. Zou et al. [14] reported that, after adding GuaSCN into a lead precursor to prepare perovskite films, the high-power conversion efficiency of solar cells was increased from 15.24% to 16.92%. According to that study, thiocyanate (SCN) ion additive significantly affects the nucleation process of perovskite film and improves film morphology; however, SCN ions are absent from the resulting perovskite film. So far, the existence of SCN groups in the final perovskite film is still debatable [15]. The improvement mechanism of SCN groups in perovskite film has also not been very clear until now. Few papers have reported that GA is not incorporated into perovskite lattice due to its large size and resides as a passivating agent. Y. Yang and co-workers [5] claimed that the GA^+^ did not substitute the methylammonium (MA) ions and demonstrated a passivation effect with perovskite film. Hou et al. [16] also reported GA^+^ could not substitute the MA ions when GCl was added into MAPbI_3_ at the ratio of less than 0.4, and GA^+^ could cross-link the neighboring gains of perovskite. The effect of pyrazine as an additive has also been demonstrated in different studies. Lee and co-workers [17] reported the incorporation of the SnF_2_-pyrazine complex to enhance the properties of formamidinium tin iodide (FASnI_3_) perovskite. The presence of excess SnF_2_ in perovskite solution facilitates phase separation. Pyrazine facilitates Sn vacancy reduction and hinders phase segregation due to the good binding affinity to SnF_2_. Solution-processed perovskite films are susceptible to higher defect densities, such as surface and grain boundaries, which significantly lowers photovoltaic performance. Hence, it is important to suppress defect formation and passivate the perovskite interface. Pyrazine acts as trapping states of charge carriers and chemically passivates surface and grain boundaries and under-coordinated Pb^2+^ defect sites in polycrystalline perovskites [18]. However, the effect of pyrazine on the photovoltaic properties of mixed Sn-Pb perovskite has not been studied widely. Organic cations are a popular and widely studied subtopic in the perovskite field, despite that fact that they bring different challenges. Due to their volatile nature, organic cations are susceptible to moisture, heat, and light, which suppress their photovoltaic performance [19,20,21,22]. Hence, all-inorganic perovskites, especially with Cs as A-cation, have recently emerged as an appealing approach to improve device stability [23,24]. Another challenge associated with Pb-based perovskite comes from the insolubility of PbI_2_. This issue with insolubility results in dendritic formation in the perovskite films. Hence, for all-inorganic lead halide perovskite, major studies focus on morphological control for polycrystalline films [25,26,27,28,29]. Some studies have incorporated Cl-based additives for the development of perovskite morphology by reducing rod-like nanostructure [30,31,32]. In addition, DMSO as a solvent improves the solubility of PbI_2_ and, thus, can improve perovskite surface texture [33,34,35,36,37,38]. Partial substitution of Pb can reduce the toxicity but still can maintain the promising optoelectronic properties that come from Pb-based perovskite. In different studies, several alternative cations have been proposed which are similar in electronic configuration to Pb. Sn is considered less toxic and is expected to offer comparable characteristics to Pb, however, it suffers from the drawback of oxidation of Sn^2+^ to Sn^4+^ [39]. Another potentially attractive but less explored perovskite composition is an all-inorganic mixed Sn-Pb halide with Cs as A cation. In this paper, we have reported the fabrication of solution-processed all-inorganic Cs(Sn, Pb)I_3_ perovskite. We have studied the effect of GuaSCN and Pyrazine additives in the bandgap tunability and enhancement in stability of Cs(Sn, Pb)I_3_ perovskite. Our results show that GuaSCN helps in the formation of lower bandgap (1.2–1.5 eV) perovskites. In addition, GuaSCN-treated perovskites show a similar absorption and structural behavior trend in samples after dark exposure for 100 h at 65 °C, indicating improved stability in Sn-Pb perovskite. Pyrazine does not alter perovskite bandgap significantly; however, it facilitates oxidation reduction. We have also reported the influence of CsCl as an additive and DMSO as a solvent in suppressing dendrite formation and improving surface morphology.

## 2. Materials and Method

Cs(Sn, Pb)I_3_ perovskite films were synthesized by a solid-state reaction as follows:CsI + xSnI_2_ + yPbI_2_ = Cs(Sn_x_Pb_y_)I_3_,
where x to y ratio of 60:40 has been used.

The procedure and instruments for films fabrication and testing follow our previous work, reported by Schwartz et al. [40]. Perovskite films were fabricated via precursor-based solution processing under the atmospheric condition, as demonstrated in Figure 1. 0.8 M solution was prepared by the mixture of Cesium Iodide (0.156 g), Tin (II) Iodide (0.134 g) & Lead (II) Iodide (0.1106 g) with a molar ratio of 60:40 and 5 mol% of GuaSCN (0.0075 g) in 1.5 mL of solvent. In another set of films, 10 mol% of pyrazine (0.0107 g) was mixed in the precursor solution instead of GuaSCN. We have used equimolar of gamma-Butyrolactone (GBL) and Dimethylformamide (DMF) for solvent. In all the fabricated films, 40 mol% of Tin (II) Fluoride (0. 125 g) has been added as an additive. To study the effect of CsCl, 10 mol% of CsCl (0.002245 g) has been added to one set of films. Similarly, to study the effect of DMSO, it has been added to the solvents with an equimolar ratio, while the volume of solvent remained the same. The precursor mixture was heated at 75 °C for 1.5 h, followed by drop-casting on preheated Tec10 substrate. Doctor blade technique was used to spread the solution over the preheated substrates. Films were then annealed in a vacuum oven at −15 in Hg and 100 °C for 1 h. All the solutes were purchased from Alfa-Aesar, Haverhill, MA, USA (Cesium Iodide, Alfa Aesar CAS: 7789-17-5; Tin (II) Iodide, Alfa Aesar CAS: 10294-70-9; Lead (II) Iodide, Alfa Aesar CAS: 10101-63-0; Tin (II) Fluoride, Alfa Aesar CAS: 7783-47-3; CsCl, Alfa Aesar CAS: 7647-17-8; GuaSCN, Alfa Aesar CAS: 593-84-0; Pyrazine, Alfa Aesar CAS: 290-37-9), and solvents from Sigma-Aldrich, St. Louis, MO, USA (GBL, Sigma-Aldrich CAS: 96-48-0; DMF, Sigma-Aldrich CAS: 68-12-2; DMSO, Sigma-Aldrich CAS: 67-68-5). Films were stored in a nitrogen-filled glove box before testing. X-ray diffraction spectroscopy (XRD) measurements were conducted on a Bruker diffractometer from Bruker Corporation Billerica, MA, USA under ambient conditions using Cu Kα radiation. EVA toolbox and Topas were used for XRD data analysis and phase identification. Shimadzu UV-2600 spectrometer was used to perform optical transmittance and reflectance measurements. The optical bandgap of the samples was determined by Tauc analysis. ATLAS SUNTEST XXL from Ametek Berwyn PA was used to conduct heat and light soaking (HLS) stability tests at 65 °C and AM1.5G solar spectrum for 100 h. JEOL JSM-5610 was used to perform Scanning Electron Microscopy (SEM) and Energy Dispersive X-ray Spectroscopy (EDS) analysis.

## 3. Results and Discussions

Figure 2a,b show the crystal structure of CsSnI_3_ with the *Pnam* space group and the crystal structure for orthorhombic δ-phase of CsPbI_3_ with Pmnb space group, respectively. The δ-phase of CsPbI_3_ has a wider bandgap (2.8 eV) that might be responsible for the wider sub-bandgap in our films shown in the following section.

Marco et al. [5] reported a bandgap of MA-based perovskites to remain unaffected with GA incorporation. According to that study, due to the bigger size of GA (278 pm) than MA (217 pm), GA cannot substitute for MA within the perovskite crystal lattice, suggesting GA cannot form a 3-D perovskite as a sole A cation. Other studies show bandgaps are relatively unaffected when GA-based additives are used, such as guanidinium isothiocyanate GITC [14], GCl [16], and GTC [13]. On the other hand, our studies show a significantly reduced bandgap from 2.5 eV to 1.5 eV with the incorporation of 5 mol% of GuaSCN, as depicted in Figure 3a. The concentration of GuaSCN is critical in determining whether the nucleation density or growth of perovskite is affected or unaffected with GuaSCN inclusion [10]. Some studies suggest a high concentration of GA-based additives can partially incorporate into perovskite lattice [13,14,16]. Our result with lowered bandgap suggests GuaSCN might partially enter perovskite lattice, which is in good alignment with this suggestion. However, sub-bandgap states can be present in the GuaSCN-added sample [13]. In our study, the sub-bandgap at around 2.45 eV with GuaSCN, which is close enough to the no-additive sample, can be ascribed to the fact that GA incorporation does not alter bandgap. As shown in Figure 3b, within the range of 300–500 nm, the absorption pattern was unaffected; however, in the visible range of 500–800 nm, absorption increased significantly with different absorption tails, suggesting a passivating effect of GuaSCN in suppressing non-radiative trap centers and improving charge transfer. The unaffected region of absorbance at a lower wavelength may be ascribed to the sub-bandgap of 2.4 eV. We repeated a similar experiment with pyrazine additive. Our study shows the pyrazine-assisted film has a bandgap close to 2 eV, which is prominently lower than the non-additive film but wider than the GuaSCN-assisted film (Figure 3a). In addition, the absorption in the 500–1000 nm range is noticeably lower than the GuaSCN film (Figure 3b). Slightly red-shifted absorption opening in the lower wavelength in pyrazine-assisted films may be ascribed to the absence of a sub-bandgap, which we observe in the case of GuaSCN film. GuaSCN also assisted in darkening the color of the final perovskite film. After 1.5 h of heating, perovskite solution with pyrazine and no additive was yellow in color; however, GuaSCN-induced solution turned black after a few minutes of heating, suggesting GuaSCN assists in faster perovskite formation, as shown in Figure 1.

The x-ray diffraction (XRD) pattern, both with and without GuaSCN, shows the mixed phase of CsSnI_3_ and CsPbI_3_ implying intercalation of PbI_2_ and formation of double B-cation Cs(Sn, Pb)I_3_ perovskite, as shown in Figure 4a. However, the GuaSCN samples have a preferred orientation, while few perovskite peaks disappear in no-GuaSCN samples confirming phase separation or incomplete perovskite formation. The significantly higher intensity with GuaSCN samples confirms better crystallinity has been achieved, and growth of perovskite crystals was structurally influenced by the addition of GuaSCN. The peaks at around 26° and 27° correspond to the (121) and (221) facets of perovskite, respectively. With the addition of GuaSCN, these peaks have shifted to a lower 2-theta angle, suggesting partial incorporation of GA in perovskite lattice. The cell volume of GA and SCN ion is larger than Cs and I ion, respectively. Thus, this peak shift to lower angles may result from the partial substitution of Cs ions with GA ions and I ions with SCN ions, similar to what was reported by Zou et al. [14]. The peaks at around 23° and 29°, corresponding to the (120) and (032) facets, respectively, arise from the SnO_2_ emerging in samples with no additives, implying additive-assisted films to have a lower oxidation state.

Our SEM images in Figure 4b–d show all the samples, with and without additives, have dendritic formations. Some studies have reported the presence of similar solid fiber in Cs-Pb perovskite [22,26,27,41]. In our study, we suggest this dendrite formation comes from the coagulation tendency of Pb-compound due to the insolubility of PbI_2_. Higher atomic % of oxide in non-additive samples found in our Energy Dispersive X-ray Spectroscopy (EDS) spectra is in good alignment with our XRD result. EDS spectra collected in several regions show the atomic weight of oxide is more than 50% in the non-additive samples which is significantly higher than in GuaSCN and pyrazine-added samples.

### 3.1. Effect of Cesium Chloride

Some studies have incorporated chloride-based additives to improve perovskite morphology by reducing the solid fiber-like characters [30,31,32]. In our study, CsCl addition in perovskite precursor diminishes isolated needle-like dendrites and improves surface coverage, as depicted in Figure 5a, compared to the GuaSCN film, as shown in Figure 5b. CsCl addition facilitates the morphology by producing interconnected grains. The average grain size increment is significantly visible with CsCl addition, and similar results have been reported with the addition of chloride-based additives in other studies [42,43,44]. The intensity of the absorption band is higher with CsCl in the visible range of 300–500 nm. However, the absorption tail is blue shifted, as shown in Figure 5c, consistent with previous works with Cl-doping [42,45]. A similar trend has been observed in the Tauc plot with blue-shifted sub-bandgap in the CsCl-added sample, as shown in Figure 5d. Some studies show the amount of Cl-ions reduces with the annealing process [45,46]. Herein, we have reduced annealing time from 60 min to 10 min to facilitate the effect of Cl-ions in the formation of the perovskite. Samples with modified annealing time show significantly higher absorption, and the sub-bandgaps were noticeably weakened, confirming the evolution of nearly single-phase perovskite compound. The overall primary bandgap was relatively unaffected with CsCl addition, which is in good alignment with the similar perovskite orientation, in both with and without CsCl samples, presented in XRD analysis. However, the relative intensities in the XRD pattern along (121) and (221) facets shown in Figure 5e reduced with CsCl, and this result is consistent with previous work reported by Li et al. [47], suggesting enhancement in the perovskite coverage with weakened growing orientation. Though the difference in intensities often arises from instrumental modes, such as centering the sample opposite to the X-ray beam or abundance of material within a particular area, all the perovskite peaks in CsCl-assisted samples show 2-theta shift to higher angles, attributing reduced lattice parameters of the perovskite unit cell. This peak shift tendency is consistent with other studies [42,43,45] and can be attributed to the substitution of iodine ions with chlorine ions.

### 3.2. Effect of Dimethyl Sulfoxide

DMSO is a popular polar solvent that has been widely used to improve perovskite solubility. Due to its strong coordination with PbI_2_, DMSO facilitates dissolution of this compound. DMSO in the solvent dissolves the PbI_2_ and SnI_2_ molecular groups in the precursor to a smaller size, resulting in smaller crystal grains during film formation [33,34]. DMSO also facilitates microstructure control in Sn-Pb alloyed perovskite films, thus preventing colloidal agglomeration and inhomogeneous Sn and Pb distribution [34]. The needle-like dendrites in our GuaSCN sample completely disappear with the addition of DMSO mixed with GBL and DMF with an equimolar ratio, as shown in Figure 5f. Our EDS result shows almost equimolar Sn/Pb with an average ratio of 1.02:1 and homogeneous distribution of PbI_2_ has been achieved with mixed DMSO, while, in GuaSCN films, the average ratio of Sn and Pb is approximately 1.8:1.

Due to the weaker bonding capacity and higher evaporation rate of DMF, the formation of perovskite from precursor solution is faster [35], and the same goes for the GBL. That is why we have observed retardation in the faster formation of perovskite with the addition of DMSO in the precursor as being a lesser volatile solvent than DMF and GBL. In our initial precursor solution with mixed DMF:GBL (1:1:1), black color appears just a few minutes after heating the solution; however, with mixed DMSO (GBL:DMF:DMSO = 1:1:1), this process takes a significantly longer amount of time. So, we can say the addition of DMSO slows down the reaction rate of perovskite growth.

Sn-Pb perovskites with mixed DMSO have the characteristic of widening bandgap. The introduction of more Pb into the perovskite lattice can result in a wider bandgap with a gradual blueshift in absorption onset. The bandgap of <1.5 eV in the GuaSCN sample is increased to 1.9 eV in the mixed-DMSO sample.

By working as a strong capping agent for SnI_2_ and PbI_2_, DMSO can retard the crystallization of Sn-Pb mixed perovskite [37,38]. In addition, the slow reaction rate of DMSO leads to poor perovskite crystallinity and random orientation [35]. Our XRD result is in good alignment with these previous studies. DMSO added films have a different diffraction pattern than GuaSCN films. The characteristics peaks at 25.7°, 26.5°, and 27.2° have a different orientation in the GuaSCN sample. The peak at the (222) plane has a significantly lower intensity that can be ascribed to the poor evaporation rate of DMSO, which can complicate the arrangement of the crystals [36].

### 3.3. Stability of GuaSCN Sample

Since bandgap tunability of GuaSCN additive led us to achieve the preferred lower bandgap, we moved forward with GuaSCN films to run light and dark cycling stress tests. After more than 100 h of the dark stress test at 65 °C, films maintained the bandgap and absorption pattern of the initial as-deposited films; however, a slight decrease in bandgap with a slight increment in absorption has been observed, confirming dark aging has improved the optical properties, according to Figure 6a,b. Similar orientation in XRD pattern and comparable intensity were also preserved in films after dark testing, shown in Figure 6c. A slight shift to higher angles has been observed that can be correlated to reduced lattice parameters. The presence of dendrites is still visible in dark stressed films; however, reduction in the size of these dendrites and improvement in morphology with compact coverage are noticeable. These results suggest dark aging facilitates completing the perovskite transition reaction, confirming our developed film with GuaSCN is highly stable under the dark stress test. However, the absorption pattern shows a major blueshift and wider bandgap of <2.5 eV after films are exposed to light stress test. The XRD shows a similar pattern, however, with lower intensity and disappearance of few perovskite peaks. EDS data shows an approximate increment in Sn-concentration from 11.233% in as-deposited GuaSCN films to 24.375% in light-exposed films that may result in the generation of SnO_2_ vacancy. The evolution of Sn-vacancy detrimentally affects perovskite stability.

A similar trend has been observed in the stability of the films with pyrazine and no additive. However, the enhancement in optical properties we observe in dark exposed GuaSCN sample is missing with pyrazine and no additive. Rather, blueshift in absorption and a slightly wider bandgap trend have been observed in dark aging samples. Considering these results, we can conclude that GuaSCN-induced films have better stability in the dark stress test.

## 4. Conclusions

In conclusion, Cs(Sn, Pb)l_3_ films were prepared by solution processing, and the effect of different additives and annealing conditions on the properties of films were studied. As deposited, non-additive films show a wide bandgap. Unlike pyrazine, the addition of GuaSCN lowers the bandgap significantly to 1.5 eV and enhances the absorption coefficient. The highly oriented mixed phase of CsSnI_3_ and CsPbI_3_ has been observed in XRD, suggesting the formation of Cs(Sn, Pb)I_3_ perovskite. Both GuaSCN and Pyrazine facilitate oxidation reduction; however, the non-additive films are prone to high oxidation, as confirmed by EDS analysis. To mitigate the elongated large crystal plates that emerged from PbI_2_, we introduced CsCl into the precursor. More intercalation of PbI_2_ into the grains led to the formation of Cs(Sn, Pb)I_3_ with few regions of white dendrites. This effect of large dendrites has been completely removed when DMSO was mixed as one of the solvents in the GuaSCN precursor. The homogeneous concentration of Sn-Pb has been achieved as DMSO facilitates solubility of perovskite solution. The absorbance in the visible range of 300–500 nm is strongly altered with CsCl addition, resulting in higher absorption. This set of CsCl added films have been further optically improved with higher absorption with a shift to higher 2-theta values, when annealing time was lowered from 1 h to 10 min. Our stability study shows fabricated Cs(Sn, Pb)I_3_ films are highly stable after dark exposure and retain the bandgap and crystal structure. An increase in absorption coefficient with a shift to higher 2-theta values suggests dark exposure completes the perovskite formation reaction. This study suggests the addition of GuaSCN additive is a potential route to enhance optical properties and stability of perovskite films when paired with reactive agents, such as CsCl and DMSO. Different fabrication approaches can be undertaken, or reactive agents can be employed, to produce films lesser prone to degradation under light exposure.

## Figures and Tables

**Figure 1 materials-15-00899-f001:**
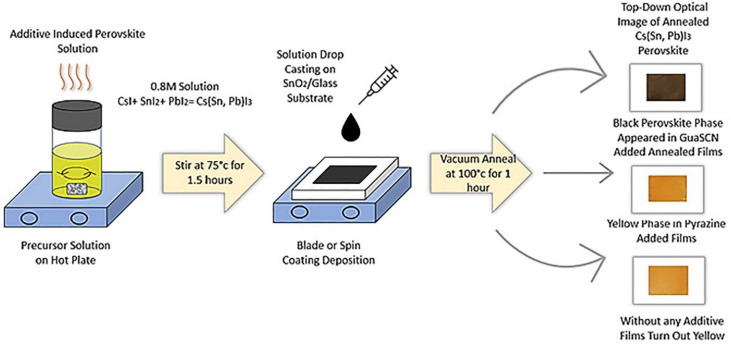
Chemical synthesis and solution processing of Cs(Sn, Pb)I_3_ perovskite films with and without additives.

**Figure 2 materials-15-00899-f002:**
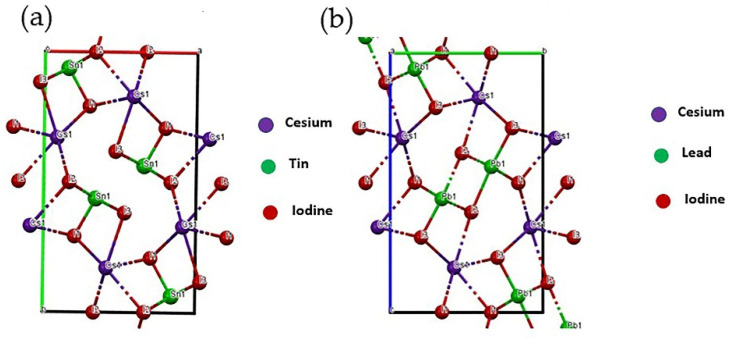
(**a**) Crystal structure of CsSnI_3_ (ICSD 14070) with Pnam space group, lattice parameters: a(Å) = 10.346, b(Å) = 17.656, c(Å) = 4.741 with Cesium, Tin, and Iodine; (**b**) crystal structure of orthorhombic δ-CsPbI_3_ (ICSD 27979) with Pmnb space group, lattice parameters: a(Å) = 4.788, b(Å) = 10.432, c(Å) = 17.758 with Cesium, Lead, and Iodine. Both the figures are reproduced in Mercury software.

**Figure 3 materials-15-00899-f003:**
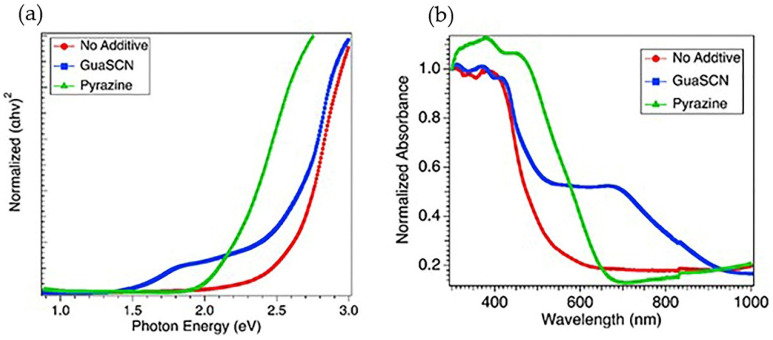
(**a**) Direct bandgap spectra shown in Tauc plot for GuaSCN film (blue), pyrazine added film (green), and film with no additive (red); (**b**) Uv-Vis absorption spectra for similar types of films represented in (**a**).

**Figure 4 materials-15-00899-f004:**
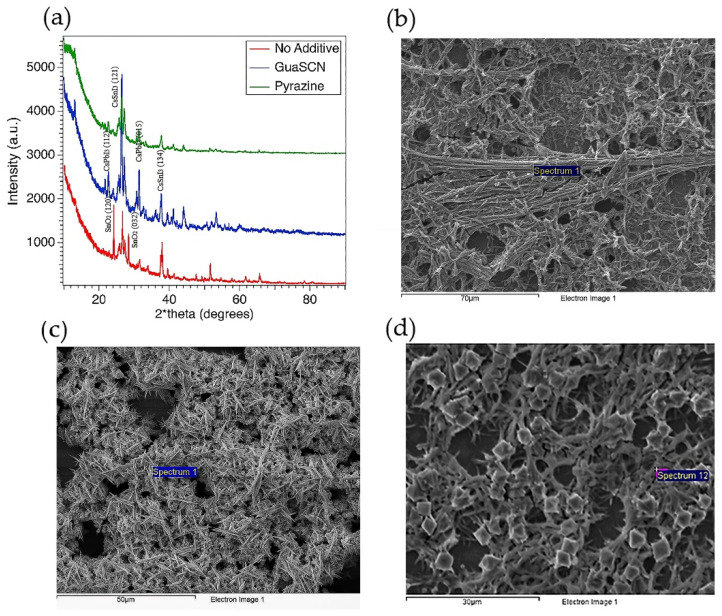
(**a**) XRD pattern for mixed Sn-Pb films with GuaSCN (blue), pyrazine (green), and no additive (red). Y-axis represents X-ray intensity in counts per second (CPS); top-down SEM image of GuaSCN film (**b**), pyrazine added films (**c**), and films with no additive (**d**).

**Figure 5 materials-15-00899-f005:**
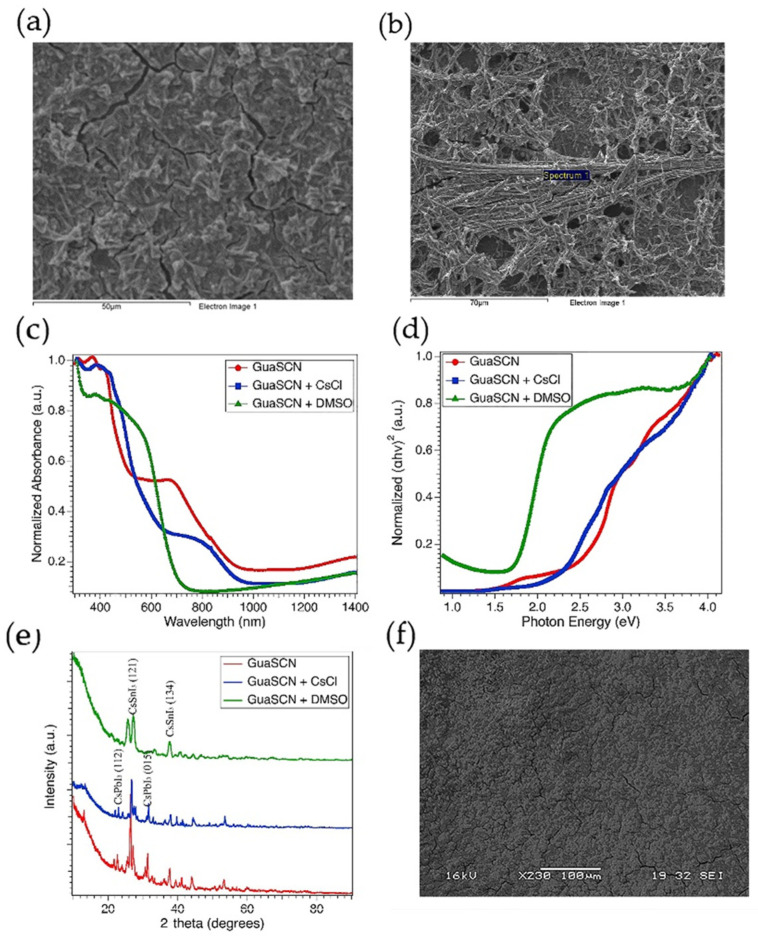
(**a**) Direct bandgap spectra shown in Tauc plot for GuaSCN film (red), when CsCl was added (blue), and when DMSO was used with other solvents (green); (**b**) Uv-Vis absorption spectra for similar types of films shown in (**a**); (**c**) XRD spectra with and without CsCl and DMSO; top-down SEM image of GuaSCN film (**d**), GuaSCN and CsCl combined (**e**), and GuaSCN and DMSO combined (**f**).

**Figure 6 materials-15-00899-f006:**
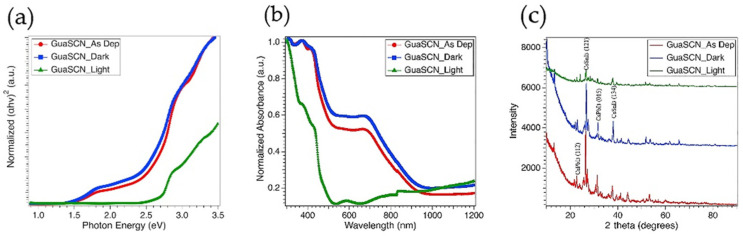
(**a**) Direct bandgap spectra shown in Tauc plot for as-deposited film (red), dark exposed film (blue), and light-exposed film (green). All of these are GuaSCN added films; (**b**) Uv-Vis absorption spectra; (**c**) XRP pattern for mixed Sn-Pb films with GuaSCN before and after stability testing. Y-axis represents X-ray intensity in counts per second (CPS).

## Data Availability

Any data requests should be sent to the corresponding author.

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
