# Peer review of "Additive-Assisted Optimization in Morphology and Optoelectronic Properties of Inorganic Mixed Sn-Pb Halide Perovskites"

_materials, 2022, doi:10.3390/ma15030899_

Round 1
Reviewer 1 Report
The manuscript is well written and presented in a logical manner. However there are some minor issues to be addressed before the acceptance of the manuscript for its publication:
- In abstract, please remove first four lines as it is like introduction part.
- Introduction part must be elaborate with some new references.
- The authors should indicate the importance of additives which were used in this study.
- In introduction, figure 1 should be placed after one or two paragraph.
- Please check all errors throughout the full manuscript in terms of grammatical mistakes, typo mistakes, space problems etc. like in line 96, 296 line 126, small letters should be in line 127 and 128, line 148 , line 177 sub script problems etc.
- Section 3 Result should Results and discussion.
- Section 3.2. please use the full in main heading.
- Please provide high quality figures because all are blurred form.
- It is necessary to indicate the significance of this review more clearly in conclusion section.
Reviewer 2 Report
Dear Authors.
Please answer the following questions and do the necessary changes.
Move Lines: 111-115 and Figure 1 from Introduction section to Materials and Method section.
Results and Discussion: The paper has clear explanation about results obtained from an experiments with proper cross citations.
Conclusion: Line: 294, repeating words "suggesting". Apart from that conclusions are well-written.
In the manuscript, there are many grammatical errors and citation issues. Please cross-check everything through out the manuscript.
Also, the graphical representations such as comparison plot and SEM images are in low-quality. Try to improvise in whole manuscript.
Please do the changes and resubmit the manuscript.
Reviewer 3 Report
The manuscript prepared by Murshed and coworkers studied the additive-assisted optimization in the preparation of mixed Sn-Pb inorganic halide perovskite films. This work is well done, and thus, I recommend the publication of this paper at journal Materials. However, before the acceptance of this submission, few suggestions may be considered by the authors.
- 3a, it does not seem like the XRD pattern has the background subtracted. This seems to be same case for Fig. 4c.
- Authors should add a scale bar for Fig. 3c and Fig. 4e.
- Line 195-196, please provide your EDS data so that one can better see the agreement between XRD and EDS results.
- Line 237-239, please provide your EDS data to show that almost equimolar ratio of Sn/Pb with homogeneous distribution of PbI2 was achieved.
- Similarly for line 281-282, please provide your EDS data to show the increment in Sn-concentration.
- Line 303, what does the author mean in terms of ‘lower annealing time collects electron more efficiently’?
Round 2
Reviewer 2 Report
Thank you for considering and modifying my comments.
The paper can be accepted as it is.
Thank You.